# Prospect and Challenges for Sustainable Management of Climate Change-Associated Stresses to Soil and Plant Health by Beneficial Rhizobacteria

**Aniruddha Sarker [1], Most. Waheda Rahman Ansary [2], Mohammad Nabil Hossain [3] and Tofazzal Islam [4,\***

1. School of Applied Biosciences, College of Agriculture and Life Sciences, Kyungpook National University, Daegu 41566, Korea; fagunaniruddha@gmail.com
2. Department of Convergence Study on the Ocean Science and Technology, Korea Maritime and Ocean University, Busan 49112, Korea; waheda.rahman@yahoo.com
3. College of Chemistry, Chemical Engineering and Biotechnology, Donghua University, Shanghai 201620, China; marchnabil@gmail.com
4. Institute of Biotechnology and Genetic Engineering (IBGE), Bangabandhu Sheikh Mujibur Rahman Agricultural University, Gazipur 1706, Bangladesh
* Correspondence: tofazzalislam@bsmrau.edu.bd; Tel.: +88-01-7140-014-14

**Abstract:** Climate change imposes biotic and abiotic stresses on soil and plant health all across the planet. Beneficial rhizobacterial genera, such as *Bacillus, Pseudomonas, Paraburkholderia, Rhizobium, Serratia*, and others, are gaining popularity due to their ability to provide simultaneous nutrition and protection of plants in adverse climatic conditions. Plant growth-promoting rhizobacteria are known to boost soil and plant health through a variety of direct and indirect mechanisms. However, various issues limit the wider commercialization of bacterial biostimulants, such as variable performance in different environmental conditions, poor shelf-life, application challenges, and our poor understanding on complex mechanisms of their interactions with plants and environment. This study focused on detecting the most recent findings on the improvement of plant and soil health under a stressful environment by the application of beneficial rhizobacteria. For a critical and systematic review story, we conducted a non-exhaustive but rigorous literature survey to assemble the most relevant literature (sorting of a total of 236 out of 300 articles produced from the search). In addition, a critical discussion deciphering the major challenges for the commercialization of these bioagents as biofertilizer, biostimulants, and biopesticides was undertaken to unlock the prospective research avenues and wider application of these natural resources. The advancement of biotechnological tools may help to enhance the sustainable use of bacterial biostimulants in agriculture. The perspective of biostimulants is also systematically evaluated for a better understanding of the molecular crosstalk between plants and beneficial bacteria in the changing climate towards sustainable soil and plant health.

**Keywords:** rhizobacteria; biostimulants; plant stress mitigation; genome editing; climate change

## 1. Introduction

Extensive nutrient mining and risky application of synthetic agrochemicals, including various chemical fertilizers and growth regulators, are considered the triggering factors for the deleterious fate of global arable soils in conventional farming [1]. Therefore, sustainable nutrient cycling and soil health management become challenging through conventional farming practices. To address these critical complications, soil-inhabiting rhizobacteria with beneficial features for plant growth were regarded as the potential alternatives for synthetic fertilizers [2]. Beneficial rhizobacteria should be considered as a sustainable option instead of conventional practices due to multifunctional benefits in terms of cost, environmental impact, and soil fertility [3]. However, the sustainable application of these

potential and beneficial rhizobacteria was considered as the hammer and nails of green agricultural microbiology.

Rhizobacteria are bacteria from diverse genera which are living in the rhizosphere of plants. Among the rhizobacterial bulk, several genera of bacteria have been proven for plant growth promotion are termed as 'beneficial rhizobacteria' [4,5]. There are various modes of action for plant growth promotion through the adoption of beneficial rhizobacteria [6]. The growth promotion should be rendered by direct mechanisms, such as nutrient cycling and solubilization, production of phytohormones, and modulation of bioactive materials [5–11]. In addition, several indirect mechanisms, such as biocontrol activities, induction of stress mitigating genes, production of secondary metabolites, and volatile organic compounds via beneficial rhizobacteria, were reported during root colonization [8,11,12]. Several potential genera, such as *Bacillus, Pseudomonas, Enterobacter, Lysobacter, Serratia*, and *Burkholderia*, have exhibited excellent growth promotion and plant defense features towards sustainable agriculture through genetic advancement of soil-inhabiting beneficial rhizobacteria [7–10]. However, *Bacilli* are considered as the key drivers for the enhancement of plant growth through biostimulation and biocontrol mechanisms [10,11]. As a result, beneficial rhizobacteria present innate potential for plant nutrition maintenance.

In general, a plant can modulate its physiological strategies to tolerate various climate change-oriented abiotic stress conditions, such as drought, salinity, heat stress, and metal toxicity [5,6,11]. However, osmoregulation, proline accumulation, total soluble solid and osmotic adjustment, antioxidant activities, regulation of stress-responsive gene (SOD, GR, CAT, POD, etc.), and inducing heat shock proteins were considered as predominating mechanisms of plants to ameliorate the stressful conditions [6,7]. Thus, there is an urgent need to introduce an environment-friendly approach for sustainable agriculture to feed the growing population worldwide [13]. Beneficial rhizobacteria may enhance crop productivity and improve plant growth by handling the stressful conditions of plants, such as plant diseases, pest attacks, and various biotic and abiotic stresses in a sustainable manner [13,14]. Increasing evidence of earlier research revealed that beneficial rhizobacteria could enhance soil fertility, nutrient bioavailability, and plant growth and development while maintaining the surrounding environment in an ecofriendly manner [15,16].

Rhizobacteria enhance a plant's tolerance to various stresses, such as pest infestation, drought, salinity, hot and cold stresses, and improve the yield of crops under changing climatic conditions [17,18]. Despite the multifunctional benefits of rhizobacteria for sustainable plant nutrient management, there are still several challenges that exist as a barrier to the commercial application of these rhizobacteria in real field conditions [13,19,20]. Thus, soil microbiological research has extended to the elucidation of the mechanisms for the prevailing constraints of the commercialization of beneficial rhizobacteria [14]. In light of the current research demand for beneficial rhizobacteria and their long-term application in soil and plant health, the goal of this review is to focus on current beneficial rhizobacteria research trends, existing research uncertainties, and practical challenges for commercial and field applications. The specific goals of this review were to (i) explore the multivariate features of beneficial rhizobacteria for mitigating climate change-related stress conditions; (ii) accelerate the understanding of the underlying mechanisms of beneficial rhizobacteria-mediated stress mitigations; and (iii) unlock the potentialities of biotechnological advancement (including OMICS-derived bacteriological engineering). Furthermore, the commercial and long-term usage of beneficial rhizobacteria for sustainable agriculture is emphasized.

## 2. Rhizosphere as a Crucial Hotbed for Plant-Beneficial Rhizobacteria Interaction

The rhizosphere is the 'playhouse' near the soil and root zone where soil, plants, and microorganisms are interlinked for the development of a platform for soil–plant–microbe interactions [20,21]. The most important stakeholders in the rhizosphere can merge their interactive physics, chemistry, and biology for making the rhizosphere a home for microscopic soil drivers, such as bacteria, fungi, and archaea [12]. The term

'Rhizosphere' was introduced by eminent scientist Hiltner in 1904. According to him, the soil zone near the root is the rhizosphere, which is distinctly divided into ecto-rhizosphere, rhizoplane, and endo-rhizosphere regions [22].

The relationship of rhizospheric bacteria and plant roots is transient and depended on the exchange of nutrients and carbon sources among the interactive organisms. 'Root exudates' are the driving force for the enhanced interaction of rhizobacteria and plant roots in the rhizosphere zone [23]. The nutrient compounds, such as carbohydrates, organic acids, and hormones present in root exudates may act as the signaling chemicals to colonize in the rhizospheric root for the soil-inhabiting microbes including beneficial rhizobacteria [8,19,22]. Thus, a mutual exchange of root exudates and plant nutrients derived from beneficial bacteria turn the rhizosphere into a hotbed for diverse genera of beneficial rhizobacteria [23] (Figure 1). However, the beneficial and positive interaction of plant roots and associated rhizobacteria primarily should be governed by the modulation of root exudates, compounds such as carbohydrates, and the flow of root exudates irrespective of plant species [24,25]. Therefore, the proliferation of the root systems is not considered as a triggering factor for plant–microbe interactions if the complex interactions and composition of root exudates are relatively poor. This kind of positive interaction of beneficial rhizobacteria and plant roots is a blessing for green and sustainable agriculture [19].

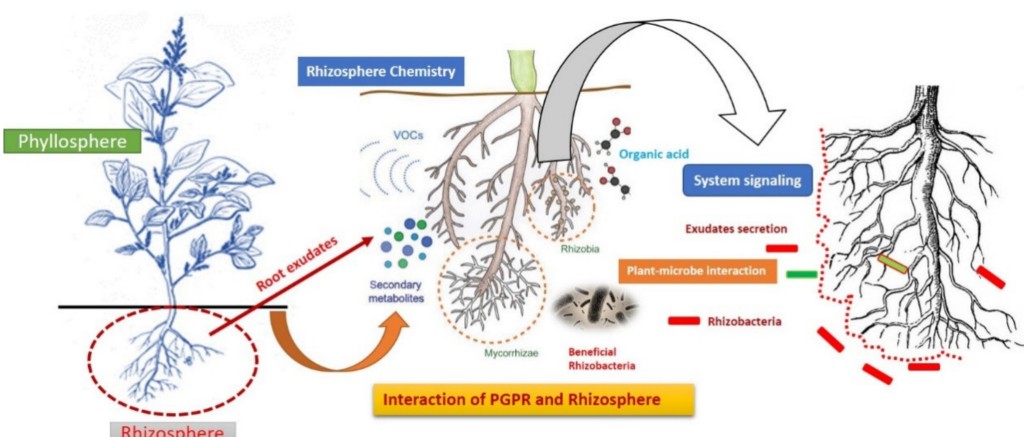

**Figure 1.** Plant-microbes interactions in the rhizosphere (a hotbed for interlinked microbes and plant root zone) mediated by system signaling, exudates secretion, and colonization potentiality are directly and indirectly improve plant growth and stress tolerance.

Although the rhizosphere is a niche for several bioactive molecules, volatile organic compounds, sugars, and organic acids, the engineering of the rhizosphere can alter the chemical feature of the rhizosphere in a positive direction [26]. The complex and biochemical and metabolic processes between plants and associated rhizobacteria are mainly facilitated by carbon deposition and their utilizations [27]. The microbial signal can vary based on the plant types, soil properties, and root exudates during the interaction of plant-microbe in the region of the rhizosphere [28]. A recent study noticed a positive correlation between the abundance of rhizospheric bacteria and fungi with soil organic matter (SOC) and conducive pH conditions [29]. Additionally, the vegetation patterns of the soil rhizosphere may also trigger the microbial ecology of the rhizosphere. Similarly, soil organic amendment is reported as the key driver for the dramatic shift of the rhizobacterial community [30]. The interaction of the N-cycler bacterial community with the nematode is evident for the sustainable improvement of nutrient cycling. Thus, a multivariate interaction to explore the plant–microbe relationship has occurred in the rhizospheric region mediated by various factors. Plant–microbe interactions in the rhizosphere simultaneously may enhance the growth of plants [31] and alleviate the plant biotic and abiotic stress conditions [8,17,18]. Therefore, the rhizosphere works as a playground for the interlinked microbes including beneficial rhizobacteria [31].

## 3. Beneficial Role of Rhizobacteria for the Enhancement of Soil and Plant Health

The soil bacteria that help to encourage plant growth while staying around the rhizosphere of plants are broadly known as beneficial rhizobacteria. The alternate name of beneficial rhizobacteria is plant growth promoting rhizobacteria (PGPR). According to previous research, PGPR has a beneficial effect on the enhancement of plant growth and sustainable plant nutrition management through direct or indirect approaches [19,29]. Several plant–microbe interactions near the plant rhizosphere are the key regulator for plant growth enhancement. However, rhizosphere chemistry is considered a triggering factor to modulate sustainable plant growth by neutral and advantageous operations, mediated by both plant roots and associated microorganisms. The system signals may ignite the mechanism of plant-rhizobacteria interaction, and thus it emerges as a positive communication among the respective stakeholders through root exudates [22,24]. A recent study demonstrated the holistic aspect of PGPR by revisiting a study aimed at the long-term and practical deployment of screened PGPR strains in real-world situations [28–32]. The detailed application of beneficial rhizobacteria is described according to their functions (Table 1).

**Table 1.** Functional groups of beneficial rhizobacteria with their salient feature and mode of action toward sustainable improvement of soil health.

| Functional Group of Rhizobacteria | Salient Feature | Mode of Action | Reference |
|---|---|---|---|
| N2 fixer (Symbiotic and Non-symbiotic) (e.g., *Rhizobium, Bradyrhizobium, Sinorhizobium, Azotobacter, Azospirillum, Gluconacetobacter, Brevibacterium*, etc.) | Nitrogen fixation and cycling can be regulated by the nitrogen fixers either by symbiotic or non-symbiotic interaction. The nodules of legumes may act as a harbor for symbiotic N2 fixers, while the rhizosphere may trigger the colonization of non-symbiotic N2 fixers. | Biological Nitrogen Fixation (BNF), Symbiotic and Non-symbiotic interaction for nitrogen fixation, Auxin (IAA) production, ACC-deaminase, Siderophore, HCN, and ammonia production | [2,13,15,33–39] |
| P-solubilizer (e.g., *Pseudomonas, Bacillus, Enterobacter, Burkholderia, Klebsiella*, etc.) | P-solubilizers may mineralize/solubilize the fixed or sequestrated phosphate in the soil due to acidic/alkaline pH conditions. The prime mechanisms comprise either enzymatic mineralization or low-molecular-weight acid secretion by the P-solubilizers. In addition, P-solubilizer may enhance the growth of plants through PGP activities by hormones/bioactive molecules | Organic/Inorganic acid production, Proton extrusion, Ammonia, and H2S production, Siderophore, Direct oxidation, Enzymatic mineralization (Particularly, by phosphatases, and phytases) | [32,40–45] |
| K-solubilizer (e.g., *Pseudomonas, Enterobacter, Acidithiobacillus, Burkholderia, Paenibacillus*, etc.) | Potassium solubilizing bacteria (KSB) are the saprophytic bacteria that can play a vital role in K cycling in soil nutrition. The actual mechanism of K solubilization is not yet confirmed, however, several predicted mechanisms (e.g., acid hydrolysis) may be the reason behind the solubilization or mobilization of insoluble K in the soil. | Acid hydrolysis of potassium from the K-minerals, Chelation of K, Undefined solubilization of potassium (Not depicted) | [16,46–48] |
| Fe chelating bacteria (e.g., *Chryseobacterium, Arthrobacter*, or other siderophore secreting bacteria) | Fe chelation and availability of soluble $Fe^{3+}$ can be mediated by the siderophore-producing bacteria. These rhizobacteria can be useful to mitigate plant stress conditions. | Siderophore production, Catechol, or other phenolic secretion | [40,49,50] |
| Hormone-producing rhizobacteria (e.g., *Bacillus, Pseudomonas, Enterobacter, Azospirillum, Staphylococcus*, etc.) | Hormone-producing rhizobacteria are acting direct role to control plant growth. The rhizobacteria-mediated hormones are auxin (IAA) and its derivatives, gibberellic acid (GA), cytokinin, etc. | Hormone secretion either under normal, or stress conditions, helps in the plant metabolism directly. | [51–55] |

Numerous earlier evaluations have documented the enhanced plant growth, improved yields, root proliferation, solubilization of P (phosphorus) or K (potassium), absorption/fixation of N (nitrogen), and some essential elements by PGPR inoculations [7,32,56,57]. However, the main contribution of PGPR to agroecological systems is $N_2$ fixation, P-solubilization, antibiotic development, and secretion of related plant growth-promoting substances [8,58,59] (Figure 2). Countable encouraging studies have already shown that agronomic returns are dramatically increasing relative to the treatment of agricultural soils with PGPR inoculations [60–62]. The unlocking of the mechanism of BNF noticed the simultaneous performance of a complex enzyme called "nitrogenase" and Nif gene interactions [33,63]. Similarly, P solubilization can be triggered by enzymatic mineralization (i.e., phosphatases, phytase, and phosphonatases), or direct solubilization through organic acid secretion, proton release, and oxidations [32,40,41]. However, other nutrient cycling, such as K-solubilization and Fe-chelation, can also be governed by several beneficial rhizobacteria from diverse genera [16,46,49].

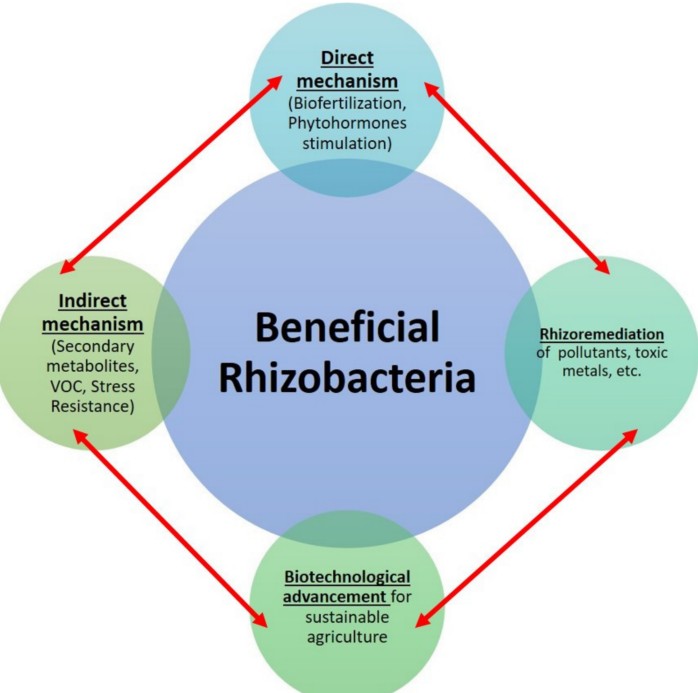

**Figure 2.** Exploration of the direct and indirect mode of action of beneficial rhizobacteria toward sustainable agriculture.

On the other hand, microbial consortia were found to be effective bioagents for plant growth promotion [64]. However, the primary function of PGPR is the promotion of plant growth as a phytostimulator or plant growth promoter and regulator [65]. Auxins, cytokines, gibberellin, abscisic acid, and ethylene are several compounds that are important to control cell processes for plant growth [61,66]. Previous studies have documented the promising impact of plant auxin (particularly indole-3-acetic acid (IAA)) could enhance cell development through enlargement and cell division, participating in major metabolic pathways to mitigate the abiotic stress condition of plants [67,68]. Although a substantial amount of research has already been reported concerning the beneficial role of rhizobacteria toward sustainable soil health, several research uncertainties still exist regarding the underlying mechanism and the interactions in the rhizosphere, which need to be investigated through applied and meticulous prospective studies.

## 4. Improvement of Plant Health under Stressful Conditions by Beneficial Rhizobacteria

### 4.1. Production of Secondary Metabolites

The production of secondary metabolites mediated by PGPR has received considerable research attention for alleviating abiotic plant stress conditions [69,70]. In the past few decades, secondary metabolites and their potential application are increasingly reported [69,71] (Figure 3). Secondary metabolites are derived from the beneficial rhizobacteria under the stress condition of plants, having an indirect effect to combat plant abiotic stress as well as support novel drug production [70]. Additionally, secondary metabolites derived from salt tolerant PGPR can be utilized for the alleviation of salinity stress in an eco-friendly manner [69]. Different PGPR strains especially *Pseudomonas* and *Bacillus* are ubiquitous and capable of producing a wide range of bioactive secondary metabolites against soil-borne pathogens [8,72–74]. The use of PGPR-derived bioactive secondary metabolites is more effective and species-specific compared to chemical pesticides [8,59]. Due to their biological origin, they are environmentally safe and have no side effects on non-target organisms [71,75].

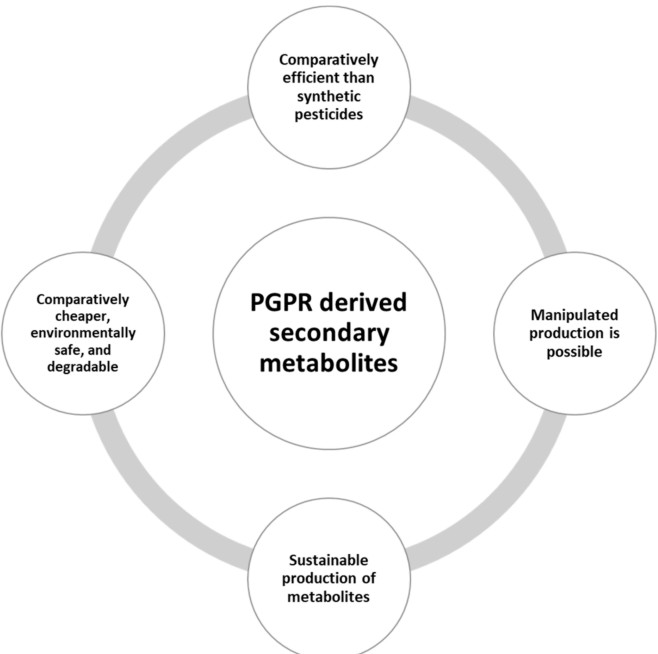

**Figure 3.** Multifunctional features of PGPR-derived secondary metabolites.

### 4.2. Enhancement of Plant Stress Tolerance

Plants are exposed to a wide range of biotic and abiotic stresses, such as extreme salinity, heat and cold stress, drought, alkalinity, and pathogenic infections [76,77]. In addition to global climate change, all these stresses are the major concern responsible for the substantial losses of crop yield, quality, and food security across global agriculture [18,78–82]. Different techniques, including the improvement of farming activities, sustainable resource management, and intensive breeding, are useful to overcome salinity, drought, or pathogenic stresses. However, they are quite costly, laborious, and time-consuming [83]. Therefore, the development of an affordable technique to alleviate plant stresses is necessary to the farmers' level. PGPR are the most contemporary and green approach and one of the best strategies for plant stress management. A large body of literature has reported that PGPR can help plants to overcome biotic and abiotic stress conditions [82,84–87]. PGPR-mediated techniques are eco-friendly, cost-effective, and sustainable than conventional techniques to mitigate plant stress conditions [75,88]. The emergence of fungal and other microbial plant diseases such as wheat blasts represents a serious threat to global food security [9]. Antibiosis is a process through which PGPRs encounter destructive pathogens through

suppressed growth and reproductive developments [8]. Those PGPRs can produce soil-friendly antibiotics. Antibiosis is the process of suppression of various diseases such as wheat blast [59,89–91], downy mildew of grapevine [92], and damping-off disease in sugar beet and spinach [8] by the bacterial metabolites. Some of these metabolites could be utilized as green pesticides for the protection of destructive plant diseases [93]. The precise modes of action of most of these bacterial antibiotics remain unknown [89,92,94].

### 4.3. Mitigation of Biotic Stress

Pathogenic bacteria, fungi, viruses, nematodes, and protists are the major biotic factors affecting agricultural production [95,96]. The PGPR can suppress the growth of phytopathogens and other deleterious microorganisms through antagonism by producing antagonistic substances (i.e., siderophores, antibiotics, hydrogen cyanide, and antimicrobial metabolites), parasitism, competing for nutrients and space, producing various lytic enzymes (e.g., chitinases, glucanases, and proteases), and inducing systemic resistance in plants [8,97–103]. The PGPR genera, such as *Bacillus*, *Serratia*, *Streptomyces*, *Pseudomonas*, *Burkholderia*, *Paenibacillus*, *Klebsiella*, *Enterobacter*, *Alcaligenes*, *Arthrobacter*, *Azospirillum*, and *Azotobacter*, are widely utilized to combat the biotic stresses [104–107]. For instance, *Pseudomonas fluorescens*, and *P. aeruginosa* are produced pyochelin and pyoverdine type siderophores to hinder the growth of phytopathogens especially the fungi by limiting the iron availability [108,109]. In addition, β-1,3-glucanase is synthesized by *Paenibacillus* and *Streptomyces* spp. and *B. cereus* to degrade the cell wall of pathogenic *Fusarium oxysporum* and some soil-derived pathogens (e.g., *Rhizoctonia solani*, *Pythium ultimum*, and *Sclerotium rolfsii*), respectively [110,111]. Moreover, plant inoculated with PGPR strain enhances resistance capacity against phytopathogens [108] reported that PGPR strains (*P. stutzeri*, *B. subtilis*, *B. amyloliquefaciens*, and *Stenotrophomonas maltophilia*) isolated from the root zone of cucumber successfully suppressed *Phytophthora capsica*, causing *Phytophthora* crown rot in cucumber through the inoculation of cucumber seeds with tested PGPR. Kanjanasopa and coworkers noticed that *Paraburkholderia* sp. strain SOS3 inhibited the growth of five fungal pathogens of rice [112]. Furthermore, rice seed inoculated with SOS3 strain improved plant growth and reduced disease symptoms of *R. solani* which cause sheath blight in rice. In addition, post-treatment with PGPR strains also responds to pathogens in several crops, including cereals (rice, wheat, and maize), chickpea, and pepper [113], *Phyllantus* [114], and tomato [115]. Therefore, the meticulous and large-scale application of PGPR is highly recommended to elucidate the mechanism of stress mitigation toward a green and sustainable future agriculture.

### 4.4. Mitigation of Abiotic Stress

The survival, growth, and development of crop plants are largely affected by abiotic stresses like salinity, high temperature, cold, and drought. They are considered as major constraints for crop productivity and causing more than 30% yield loss worldwide [18,78,116]. However, plants can grow under abiotic stresses such as drought and high temperatures supported by PGPR [67,87,117]. Accumulation of osmoprotectants, production of reactive oxygen-scavenging enzymes [catalase (CAT), ascorbate peroxidase (APX), superoxide dismutase (SOD)], glutathione and reductase), and synthesis of antioxidants (ascorbic acid, α-tocopherol, and glutathione) are the multigenic trait helps the plant to stand under abiotic stress condition [116,118]. Several studies reported that inoculation of plants with PGPR may develop a root system facilitating the uptake of more water and thereby reduce the severity of drought stress compared to the non-treated plants [118–120]. Besides, exopolysaccharides, osmolyte, stress hormones, antioxidants, plant growth regulators, and other morpho-physiological and biochemical changes aid to mitigate drought stress [121–123].

Salinity is the most destructive environmental threat that severely affects the morphological traits, physiology, and metabolic processes in plants [124,125]. The photosynthetic system collapses or reduces drastically due to salinity stress [126]. However, different

PGPR strains can effectively solve this problem. For example, PGPR influences the production of various phytohormones that improve the growth of plants, at the same time being used as an alleviator of salinity stress in plants [83,127,128]. Moreover, Rabhi et al. (2018) demonstrated that priming of *Arabidopsis thaliana* with auxin producing PGPR *P. knackmussii* MLR6 reduced oxidative damages caused by salinity stress and improved growth compared to the control plants [129]. In addition, alleviation of salt stress in rice, millets, eggplant, tomato, maize, cotton, cordgrass, and lettuce by PGPR strains, such as *Enterobacter* sp., *Xanthobacter autotrophicus*, *P. aeruginosa*, *P. fluorescens*, *P. putida*, *B. brevis*, and *Azospirillum* sp., have been reported [130–133]. Some PGPR with their potential mechanisms to ameliorate abiotic stresses are listed in Table 2.

**Table 2.** Alleviation of abiotic stresses mediated by potential beneficial rhizobacteria.

| Stress | PGPR Strain | Mechanism of Mitigation of Abiotic Stress | Beneficial Host | Reference |
|---|---|---|---|---|
| **Drought** | *P. fluorescens* (Pf1) *B. subtilis* (EPB5, EPB22, and EPB 31) | Proline accumulation, Activation of enzyme systems | Green gram (*Vigna radiata*) | [134] |
| **Drought** | *B. licheniformis* (K11) | Activation of stress-proteins and upregulation of stress-related genes (e.g., Cadhn, VA, sHSP, and CaPR-10) | *Capsicum annuum* | [135] |
| **Drought** | *Achromobacter xylosoxidans* (SF2) *B. pumilis* (SF3) and SF4) | Secretion and regulation of phytohormones. | *Helianthus annuus* | [136] |
| **Drought** | *Bacillus* spp. (KB122, KB129, KB133, and KB142) | Enhanced relative water content, improved plant physiology, retain soil moisture content, and proline contents | *Sorghum bicolor* | [83] |
| **Drought** | *B. thuringiensis* (AZP2) *P. polymyxa* (B) | Production of volatiles (e.g., benzaldehyde, β-pinene, and geranyl acetone) | *Triticum aestivum* | [137] |
| **Drought stress** | *P. aeruginosa* (GGRJ21) | Increased the levels of antioxidants, improved cell osmolytes, and upregulation of stress-responsive genes. | *V. radiata* | [138] |
| **Drought** | *Burkholderia* sp. (LD-11) | Enhanced plant physiology and growth regulators in the plants | *Zea mays* | [139] |
| **Drought** | *A. brasilense* | Alteration of physiological and biochemical changes including modulation of photosynthetic pigments, ABA, proline, and lipid peroxidation | *A. thaliana* | [122] |
| **Salinity** | *Dietzia natronolimnaea* (STR1) | ABA signaling, SOS pathway, ion transporters and antioxidant machinery | *T. aestivum* | [140] |
| **Drought and Salinity** | *B. subtilis* *A. protophormiae* (SA3) *D. natronolimnaea* (STR1) | IAA production, regulation of abscisic acid/ACC deaminase level and modulating expression of genes encoding for CTR1/DREB2 proteins | *T. aestivum* | [141] |

**Table 2.** *Cont.*

| Stress | PGPR Strain | Mechanism of Mitigation of Abiotic Stress | Beneficial Host | Reference |
|---|---|---|---|---|
| **Salinity** | *Sphingomonas* sp. (LK11) | Regulation of endogenous phytohormones (abscisic acid, salicylic acid and jasmonic acid) | *Solanum pimpinellifolium* | [61] |
| **Salinity** | *Halobacillus dabanensis* (SB-26) *Halobacillus* sp. (GSP 34) | Osmotic regulation and physiological modulation | *Oryza sativa* | [142] |
| **Salinity** | *P. putida* *Novosphingobium* sp. | Reduction of the level of ABA and SA, inhibit the proline and chloride accumulation of root | *Citrus* | [143] |
| **Salinity** | *Curtobacterium albidum* (SRV4) | Inducing systemic tolerance | *O. sativa* | [144] |
| | *B. halotolerans* *Lelliottia amnigena* | Judicious manipulation of $K^+$ and $Na^+$ uptake in roots and shoots | *T. aestivum* | [145] |
| **Salinity** | *Azotobacter* sp. (Az2 and Az6) | Improvement of physiological attributes and enhanced growth dynamics | *T. aestivum* | [146] |
| **Salinity** | *Acinetobacter bereziniae* (IG 2) *Enterobacter ludwigii* (IG 10), *Alcaligenes faecalis* (IG 27) | Modulation of chlorophyll, proline contents, total soluble sugar (TSS, electrolyte leakage, and activities of antioxidant enzymes | *Pisum sativum* | [147] |

### 4.5. Remediation of Plant Stress Caused by Pollutants

Pollutants and heavy metals are raising concerns throughout the world due to their negative effect on organisms, plants, and the ecosystem. PGPR are an eco-friendly alternative to overcome the problems of persistent organic pollutants, hazardous phenolic compounds, and toxic heavy metals [16,148]. Mechanisms, such as the production of siderophore, 1-aminocyclopropane-1-carboxylate deaminase (ACC), volatile organic compounds and phytohormones, biofilm formation, signal interference, quorum sensing, etc., are used to bioremediate the polluted soil by PGPR [148]. Rhizoremediation is the combination of phytoremediation and bio-augmentation which can be applied for better results compared to any of the single approaches to eliminate or transform pollutants from the contaminated sites [149–151]. PGPR show symbiotic and non-symbiotic relationships with plants, which is essential for rhizoremediation [152]. For example, PGPR strains *P. fluorescens* and *P. putida* were used in the study of Baharlouei et al. (2011) to protect barley plants from the toxic effect of cadmium contaminated soil [153]. Patel et al. (2016) reported that *Alcaligenes feacalis* RZS2 and *P. aeruginosa* RZS3 chelated $MnCl_2 \cdot 4H_2O$, $NiCl_2 \cdot 6H_2O$, $ZnCl_2$, $CuCl_2$, and $CoCl_2$ other than $FeCl_3 \cdot 6H_2O$ by siderophore production at batch scale, superior to chemical ion chelators EDTA and citric acid [154]. In addition, PGPR strain *Acinetobacter* sp. PDB4 degraded anthracene (low molecular weight), pyrene, and benzo(a)pyrene (PAHs, high molecular weight), has the tendency to produce biofilm, and resulted in high emulsification index under PAHs stress [155]. Therefore, bioremediation of toxic chemicals using PGPR is a more effective, suitable, and the best alternative to manage soil pollution compared to the other conventional methods [148].

### 4.6. Biocontrol Activities of Beneficial Bacteria

Nature has its mechanisms to balance the biotic community of any ecosystem. PGPR are natural, economic, eco-friendly, biodegradable, and target-specific components asso-

ciated with the plant root zone which offer protection against harmful pests directly or indirectly. The PGPR represents biocontrol properties against a wide range of soil-borne plant pathogens by producing siderophore, antibiotics, and HCN [8,59,62,156,157]. The identification and use of these beneficial microbes as biocontrol agents will reduce the use of toxic and hazardous chemical pesticides, and also decrease the problems associated with environmental pollution.

### 4.7. Production of Antibiotics and Siderophores

The production of antibiotics against phytopathogens is regarded as the most powerful biocontrol mechanism of PGPR [8,158,159]. *Pseudomonas*, *Bacillus*, *Streptomyces*, and *Rhizobium* are the most studied PGPR strains [13,160–162]. They synthesize antibiotics, such as amphisin, oomycin A, phenazine, pyoluteorin, 2,4-diacetylphloroglucinol (DAPG), tensin, pyrrolnitrin, tropolone, surfactin, iturins, bacillomycin, kanosamine, oligomycin A, zwittermicin A, and xanthobaccin that are the key weapons to defend against a wide variety of plant pathogens [8,91,110,160,163–165]. Besides, siderophores are the extracellular, low molecular weight compounds produced by the PGPR strains [166]. These organic compounds are produced under Fe-stressed conditions. Their primary function is to chelate iron from the soil solution and make it unavailable for the pathogens under limiting iron conditions [162,167]. The PGPR strains also synthesized siderophore for induction of systemic resistance gene to combat abiotic plant stress [168].

### 4.8. Induced Systemic Resistance

Crosstalk between plants and PGPRs helps to stimulate systemic resistance in the plant during the pathogenic invasion. In this process, some defense enzymes (e.g., SOD, CAT, and APX) for scavenging of reactive oxygen species (ROS), and other associated enzymes such as β-1, 3 glucanase, chitinase, and polyphenols related to plant defense are activated and help the plant to resist against pathogenic attack [169]. *Pseudomonas* induced systemic resistance in radish, *Arabidopsis*, and carnation via the "O antigenic side chain" present in the outer membrane of bacteria. A universal PGPR, *Bacilli* protect plants from the attack of notorious microbes, nematodes, and viruses by triggering the pathways of induced systemic resistance [72]. A variety of PGPR compounds like 2,3 butanediol, acetoin, acyl-homoserine lactones, cyclic lipopeptides, lipopolysaccharides, SA, and siderophores are reported to induce systemic resistance [170,171]. Additionally, protective enzymes are produced by the PGPRs to damage the cell walls of pathogenic organisms [172]. The β-1,3 glucanase and chitinase produced by *P. fluorescens* and *Sinorhizobium fredii* can break down the chitin and N-acetylglucoseamine of the fungal cell wall and thus control fungal diseases caused by *F. oxysporum* and *F. udum* [173]. It is also an effective and quick method to control plant pathogens.

## 5. Application of PGPR for Sustainable Soil Health and Native Microbial Diversity

The soil health and productivity of the soil are mostly controlled by the abundance and interaction of the native soil microbial community [174–178]. Soil-inhabiting beneficial rhizobacteria (i.e., PGPR) are the most influential and environmentally sustainable biological drivers to enhance soil quality and fertility [179–185]. Beneficial rhizobacteria may help in fixing atmospheric nitrogen, biogeochemical cycling of minor plant nutrients, solubilization of soil-bound nutrients, production of plant growth promoting hormone, and modulation of polysaccharides. These polysaccharides are helpful during soil structure formation and thus maintaining the sustainable biodiversity of soil biota and improved crop production [186]. Potential PGPR may regulate nutrient cycling through various mechanisms as an in-situ green alternative to synthetic fertilization [187,188]. In addition, PGPR improves soil health through decomposing crop residues, synthesizing and mineralizing soil organic matter [189]. Soil organic matter and root signaling for the secretion of exudates at the rhizosphere region act as the key factors for the positive interaction of the rhizospheric microbiome with the plant to improve soil quality [25,29]. Thus, PGPR are

considered the best natural alternatives to chemical fertilizers to maintain sustainable soil health.

However, potential PGPR strains isolated from the rhizosphere can influence the diversity of the indigenous rhizosphere microbiome by modifying the quality and quantity of root exudates, producing antibiotics and siderophores, and maintaining a cooperative relationship with closely associated microorganisms [102,103,190]. Besides, the addition of exogenous PGPR is regarded as a potential competitor to native microbial activities and diversities [176,191–197]. Therefore, the potential change of the microbial population affected by the PGPR is a key concern of soil microbial ecology, while PGPR are now widely used to improve crop production [198]. Kokalis-Burelle et al. (2006) reported the inoculation of *Capsicum annuum* with two PGPR strains *B. subtilis* GBO3 and *B. amyloliquefaciens* IN937a did not adversely affect the native beneficial bacterial population, whereas, increased fungal population without causing any root diseases [199]. Similarly, the inoculation of pepper seedlings with a beneficial bacteria (*B. amyloliquefaciens*) increased the native bacterial and fungal biomass in the rhizosphere of soil [200]. In contrast, none of the halo-tolerant PGPR inoculations disturb the soil microbial population in maize [140], cucumber [201], and peanut [198]. However, the bacterial community in the rhizosphere mostly differed due to the soil type and crop growth stage [202]. A recent study reported the mutual relationship between the soil microbial diversity and dynamic soil function, which may be manifested by various interactions of plant–microbes for enhancement of overall soil quality [203]. Therefore, further research should be designed on different soil types and crop growth stages to explore the effect of PGPR strains on native microbial diversity.

## 6. Industry-Laboratory Research Gap for Commercial Applications

PGPR-mediated biofertilizers are very popular, effective and drew an escalating interest compared to commercial synthetic fertilizers. However, ample use of these biofertilizers from the lab to the field face several challenges. The primary concern to develop a new biostimulant is laboratory screening and their formulations linked with the effective direct and indirect mechanisms of plant growth promotion by PGPR strains in the field. However, the effectiveness may change due to their association with other microbes. Besides, unknown and inconsistent mechanisms of PGPR may limit the application of PGPR-based biofertilizers [204]. Therefore, quality carrier material, quality control legislation, sophisticated technology, as well as trained and expert personnel are required for a successful biofertilizer production and commercialization for sustainable agriculture [205–207].

Shelf-life is a technical constraint to the development and commercialization of a biofertilizer [208,209]. Recycling is risky for the biofertilizers with short shelf life if they are not sold or used before the expiry resulting in financial loss of that marketing agency. In addition, both the marketing agency and farmers need extra care and precaution during handling and storage. Mutation of live cells used in biofertilizers may occur due to different extreme conditions [210]. Due to this short shelf life, a compatible carrier like peat, charcoal, lignite, etc. is necessary for the field application of biofertilizers. However, the unavailability of these carriers poses some technical constraints for the large-scale use of biofertilizers. The regulatory processes are quite complex, and the fees are sometimes a significant consideration affecting the commercialization of biofertilizers. Documentation procedures are also expensive, lengthy, and complicated [3,102]. Due to the lack of sufficient financial resources, potential formulations of biostimulants have been destroyed or remain unknown [210]. Transportation is also one of the major problems. Due to the lack of adequate knowledge about the advantages of biofertilizers over synthetic chemicals, the demand for biofertilizers is reduced to the farmers. Finally, biofertilizers should be selected based on their performance under field conditions with a wide range of crop varieties, diversified soil types, and environmental conditions [102]. Bioengineering, biotechnology, and multi-omics studies are time-consuming for a better understanding of PGPR commercialization [211,212].

## 7. Advanced Biotechnological Tools for Improving Beneficial Rhizobacteria

The key challenges to the commercial application of beneficial rhizobacteria are the lack of suitable formulations during storage, varied performance in real field applications, and loss of potency over time [213]. These research drawbacks can be managed by enhancing the genetic makeup of beneficial rhizobacteria through various advanced genetic tools, such as genomics, metabolomics, and proteomics (see Figure 4). Among the smart biotechnological tools, next-generation sequencing (NGS) is a fairly popular attempt to screen out the target beneficial gene or gene cluster [214]. Comparative screening of the stress mitigating gene pool (i.e., hyperosmotic stress-tolerant gene) was carried out for the proteobacteria through various biotechnological approaches by Kohler and coworkers [214]. Similarly, a partial genome sequencing of *P. polymyxa* was performed to select the potential gene of beneficial rhizobacteria for sustainable agriculture [215].

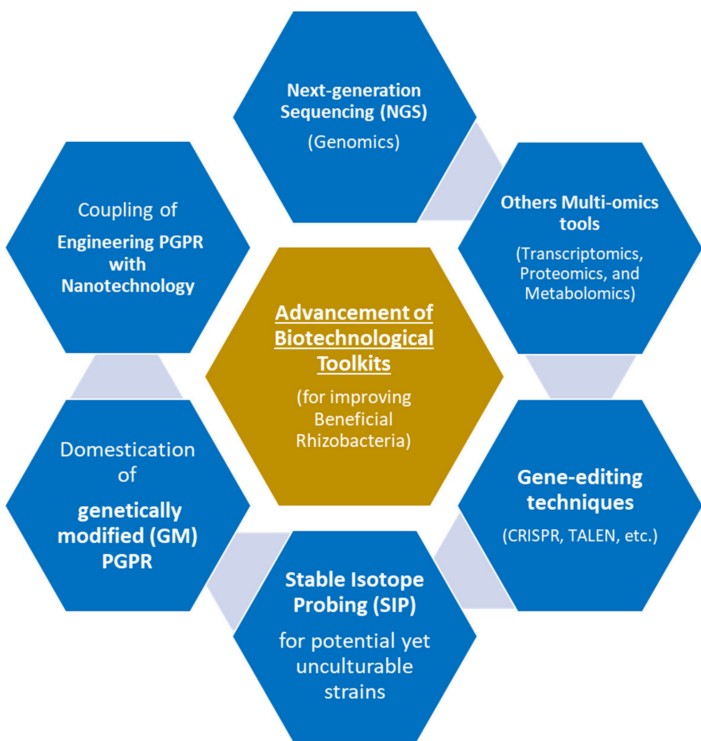

**Figure 4.** Advancement of biotechnological approaches for a sustainable enhancement of beneficial rhizobacteria.

However, an encouraging study revealed the comparative genomic feature amongst the four species of *Pseudomonas* for identifying similarities and dissimilarities of plant growth promoting and plant stress alleviating gene clusters [108]. Based on study findings, the effect of Clusters of Orthologous Groups (COGs) among the studied pseudomonads exhibited common characteristic features, such as metabolism for bioactive molecules, root colonization at the rhizospheric regions, and tolerance for salinity and toxic metals. The gene analysis through advanced genomics technique among the studied PGPR and associated proteobacteria have identified the beneficial gene sequence for the plant growth promotion [216]. Additionally, several detailed genomics studies reported that the gene contributed to plant growth promotion and enhancement of plant nutrition for *B. amyloliquefaciens* and two Streptomyces species through whole-genome sequencing [217]. Thus, advanced genomics study of beneficial rhizobacteria may expose the potential gene/gene cluster contributing to the plant growth promotion toward green and sustainable soil microbiology.

In addition, the detailed proteomics study of PGPR (*P. fluorescens*) reported the expression of the whole protein responsible for the salt stress tolerance for upregulation

of the gene and further salinity tolerance mechanism [218]. Proteomics study refers to a detailed study among the multi-omics approaches for the genetic characterization of potential rhizobacteria for plant growth promotion [218,219]. The study of proteomics should be coupled with transcriptomics to elucidate the complex mechanisms for plant growth promotion through the secondary metabolism of beneficial rhizospheric microbes [220]. However, the biggest limitation of proteomics and transcriptomics is the cross-checking of the potentiality of screened whole protein or transcriptomics, potentiality due to lack of available protein and transcriptional databases [221]. The metagenomics study for interpreting the genetic potentiality of the studied beneficial rhizobacteria from the environmental samples has a promising future in the advanced biotechnological screening of plant growth promoting rhizobacteria [222,223]. Furthermore, the innovative and interlinked studies regarding the genetic advancement of beneficial rhizobacteria can be useful for the overall development of sustainability and screening of perfect rhizobacteria for plant growth enhancement [211]. Although the application of smart biotechnological tools towards the enhancement of beneficial rhizobacteria is established, several controversial issues concerning the biosafety of PGPR studies with advanced biotechnological and microbial approaches reported [39,55,177,178,224]. Therefore, extensive and appropriate care should be taken to maintain biosafety during the molecular and biotechnological studies with potential rhizobacteria to avoid any cross-contamination and further pathogenesis. The improvement of beneficial rhizobacteria through multi-omics approaches will unveil the hidden genetic mystery towards sustainable plant growth promotion.

## 8. Omics-Driven Approaches for Engineering of Beneficial Rhizobacteria

After the emergence of next-generation sequencing (NGS), genomics became a vital engineering tool for scrutinizing target plant growth promotion (PGP) gene/gene clusters [225]. An earlier observation reported the genetic engineering of nitrogen-fixing bacteria (*P. protegens* pf-5) for enhanced biological nitrogen fixation and further sustainable application for improvement of soil N nutrition [226]. Similarly, soil phosphorus (P) nutrition can be regulated through the application of genomics to hasten the release of labile P (plant-available P) from phytate through the engineering of *Pseudomonas* sp. [227]. Furthermore, a genome-wide investigation of rice documented the potential interaction of gene clusters of rice for plant growth promoting features to combat abiotic stress conditions [228]. An admirable review noticed the multi-omics (such as genomics, transcriptomics, proteomics, and metabolomics) engineering toolkits and their holistic interaction for enhanced system biology and gene editing for screening target genes of plant growth-promoting microbes (PGPM) to harness phytoremediation [229].

Whole-genome sequencing was reported for a potential plant growth promoting bacteria (*S. marcescens* BTL07) derived from the rhizoplane of *C. annuum* to explore the target gene and genomic map for further phylogenetic profiling through advanced biotechnological tactics [230]. The molecular strategies of plant growth promoting *Bacilli* were unveiled through prospective genomics and post-genomics strategies for ensuring a sustainable and practical field application of these potent Bacilli strains [10,98,231]. Among the gene-editing tools, CRISPR is a comparatively affordable and highly encouraging gene-editing approach for the advancement of genetic engineering of rhizospheric microbiomes including beneficial rhizobacteria [232]. However, the interlinked and cross-linked omics-based engineering of beneficial rhizobacteria toward sustainable agriculture was reviewed [55,176–178,183–185,225]. A vital pitfall of beneficial rhizobacteria application in the field condition is the inconsistent performance due to the lack of optimal formulation. This drawback of beneficial rhizobacteria was minimized through the immobilization of rhizobacterial strains onto nanofibers during the seed inoculation of soybean [233]. Thus, further quorum sensing, multi-omics gene editing approaches can be coupled with advanced nanotechnology as a holistic solution to overcome the prevailing constraints of PGPR application in the real field conditions under heterogenous and adverse climatic conditions. Despite several challenges, there are countable genetically modified approaches

including CRISPR-Cas genome editing for crops and commercial biotech products concerning PGPR that are available for multifunctional applications toward green agriculture [234]. The omics-driven genetic tools can be used for the enhancement of PGPR to meet the future need for ecofriendly biofertilizers through industry–research collaboration.

## 9. Conclusions and Future Perspective

The use of beneficial rhizobacteria to improve plant growth and manage plant stress is seen as a green and environment-friendly strategy for long-term agricultural production. Beneficial rhizobacteria have been found to adopt several direct and indirect methods for improving plant health. Although research findings for helpful rhizobacteria for various plant species have been widely reported, the inconsistency of their performance in real field conditions is considered a key bottleneck for the commercial application of these elite strains of rhizobacteria. The exploration of the mode of action and prospects for the long-term use of plant growth-promoting rhizobacteria have been revealed. Furthermore, advanced research into prospective microbial consortia, formulation improvement, and a roadmap to commercial application remain in their early stages. Likewise, several methods based on biotechnological development and formulation optimization for practical use are being developed around the world to address the current limits. The present review focuses on the current trends of beneficial rhizobacteria towards the sustainable enhancement of soil and plant health, as well as strategies to combat technical hitches in the real field conditions. The advanced biotechnological tools and their application for the improvement of beneficial rhizobacteria are also discussed. Further comprehensive and interlinked studies for a better understanding of the interactions between plants and PGPR for sustainable promotion of soil and plant health by the application of these natural bioresources are needed. Recent advances in genomics, post-genomics, and genome editing toolkits would support a better understanding the molecular crosstalk between plants and the PGPR in the heterogenic climatic features.

**Author Contributions:** Conceptualization, A.S. and T.I.; data curation, M.N.H.; writing—original draft preparation, A.S., T.I. and M.N.H.; writing—review and editing, T.I. and M.W.R.A.; visualization, A.S.; supervision, T.I. All authors have read and agreed to the published version of the manuscript.

**Funding:** This research received no external funding.

**Institutional Review Board Statement:** Not applicable.

**Informed Consent Statement:** Not applicable.

**Data Availability Statement:** Not applicable.

**Acknowledgments:** Authors are grateful to the Institute of Biotechnology and Genetic Engineering, Bangabandhu Sheikh Mujibur Rahman Agricultural University, Gazipur, Bangladesh for proofreading and other technical supports.

**Conflicts of Interest:** The authors declare no conflict of interest.

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
