# Peer review of "Prospect and Challenges for Sustainable Management of Climate Change-Associated Stresses to Soil and Plant Health by Beneficial Rhizobacteria"

_stresses, doi:10.3390/stresses1040015_

Round 1

Reviewer 1 Report

The review entitled-Prospect and challenges for sustainable management of climate  change-associated stresses to soil and plant health by beneficial rhizobacteria” has the potential to be an important contribution. Overall this review study cover interesting topic, mitigation of climate change-associated stress through rhizopshere and fit in the aim and scope of the journal. I recognize that the authors have invested a lot of work with literature review and summarizing the information, and I recommend this study for further consideration.  Here I share a few ideas about how to improve the manuscript.  

Abstract is too weak and required to be improved and include quantitative finding of the review.

The originality and novelty of the paper need to be further clarified in the Introduction section. The present form does not have sufficient arguments to justify the novelty of the paper.

Proofreading should be conducted to improve organization quality.

Table 1: Suggested to add below references in Table 1.

https://doi.org/10.1080/23311932.2015.1037379

https://doi.org/10.1080/00380768.2012.716750

Line no 283-284: Support sentence with recommended studies (https://doi.org/10.1080/10590501.2019.1654809 ; https://doi.org/10.1016/j.bcab.2019.101288 )

Author Response

Reply to Reviewer 1

Comments and Suggestions for Authors

The review entitled- “Prospect and challenges for sustainable management of climate  change-associated stresses to soil and plant health by beneficial rhizobacteria” has the potential to be an important contribution. Overall this review study cover interesting topic, mitigation of climate change-associated stress through rhizopshere and fit in the aim and scope of the journal. I recognize that the authors have invested a lot of work with literature review and summarizing the information, and I recommend this study for further consideration.  Here I share a few ideas about how to improve the manuscript.

Reply: Thank you for the encouraging comments and suggestion for improvement of the manuscript. To improve the general quality of the manuscript, we have carefully followed your suggestions.

Abstract is too weak and required to be improved and include quantitative finding of the review.

Reply: Many thanks for this critical and valuable comment. The abstract is revised and the review methodology is amended for a clarification of the review process. The entire abstract is rephrased to meet the focused points of our review story (see revised abstract Line 16–36).

The originality and novelty of the paper need to be further clarified in the Introduction section. The present form does not have sufficient arguments to justify the novelty of the paper.

Reply: The originality and novelty statement of this manuscript is amended with revised, and specific objectives in the revised Introduction. See specific objectives of this manuscript amended (see Line 108–124).

Proofreading should be conducted to improve organization quality.

Reply: The entire manuscript is double-checked for proofreading and overall improvement of English grammar. The changes in the revised submission are performed using the “track change” mood of MS word.

Table 1: Suggested to add below references in Table 1.

https://doi.org/10.1080/23311932.2015.1037379

https://doi.org/10.1080/00380768.2012.716750

Reply: Thank you for deciphering some excellent reference doi. We have searched the doi for obtaining the valuable points relevant to our review, and amended our revised manuscript, accordingly (see revised Table 1 amended with reference list 235, and 236, highlighted in red font color).

Line no 283-284: Support sentence with recommended studies (https://doi.org/10.1080/10590501.2019.1654809; https://doi.org/10.1016/j.bcab.2019.101288 )

Reply: Again thanks for referring some good referral papers. We have added the suggested references (see reference cite as 233, and 234 in Line 300, the list of references is highlighted in red font color).

Reviewer 2 Report

Dear Author, 

the review is interesting and conceptually well structured.However, I have some consideration to make. 

line 47-48: in my opinion I would move the sentence before the period into lines 53-59. 

line 72-75 and line 76-77: the concept is the same in the two periods.  

line 96: in my opinion the sentence is not well written. Instead of is after 1904 "to describe the soil near.."

line 172: the parenthesis is missing

line 231: Glomus and Gigaspora are not bacteria but AM fungi. 

line 237: I think that the correct  name of bacteria is Burkholderia cepacia

line 306: PGPR are and not is

line 327-328: the sentence is not clear

Throughout the text modify the bacterial species (intended as a species or as a genus) and the scientific names of the plants in italics. Also, check that the names of the bacteria (phylum and genus) are capitalized.

Finally, standardize the references since they are not all the same. For example the 1 (when there is a double name put the period and not the hyphen), the number 18 (the names go in lowercase) etc.

King regards

Author Response

Reply to Reviewer 2

Comments and Suggestions for Authors

Dear Author, 

the review is interesting and conceptually well structured. However, I have some consideration to make. 

line 47-48: in my opinion I would move the sentence before the period into lines 53-59. 

Reply: Thank you for your encouraging comment. We have rearranged the spotted section of the Introduction to enhance the clarity of that paragraph. See Line 60–84.

line 72-75 and line 76-77: the concept is the same in the two periods.

Reply: Thanks again for the critical insight. The spotted sections were corrected to reduce the potential redundancies during revision (See Line 98–104). 

line 96: in my opinion the sentence is not well written. Instead of is after 1904 "to describe the soil near.."

Reply: The marked sentence is revised, accordingly. See Line 131–133 in the revised manuscript.

line 172: the parenthesis is missing

Reply: The parenthesis is corrected. See Line 210.

line 231: Glomus and Gigaspora are not bacteria but AM fungi.

Reply: This is an accidental typo. Those AM fungi names are deleted. See Line 270.

line 237: I think that the correct  name of bacteria is Burkholderia cepacia

Reply: Thank you for spotting a crucial issue. The corrected bacteria name is Bacillus cereus in the revised submission. See Line 275.

line 306: PGPR are and not is

Reply: Thanks for noting this concern. This issue is corrected for the whole manuscript.

line 327-328: the sentence is not clear

Reply: We revised this sentence in a more simplified way for clarity. See Line 366–367.

Throughout the text modify the bacterial species (intended as a species or as a genus) and the scientific names of the plants in italics. Also, check that the names of the bacteria (phylum and genus) are capitalized.

Reply: This was also an unintentional typo. The scientific names are rechecked and corrected, thoroughly. We are intended to species of the beneficial rhizobacteria in the revised submission.

Finally, standardize the references since they are not all the same. For example the 1 (when there is a double name put the period and not the hyphen), the number 18 (the names go in lowercase) etc.

Kind regards

Reply: This is happened due to generated bibliography from the reference manager software. However, the reference lists are corrected manually following the standard guidelines of stresses.

Reviewer 3 Report

The article submitted for review meets the standards of a review article. It raises an interesting issue of stress and beneficial rhizobacteria.  Authors  focuses on the current trends of beneficial rhizobacteria toward sustainable enhancement of soil and plant health,

Moreover, all  it is supported very reliably by a large amount of literature.

All tables are presented in a clear way and also figures.

In my opinion this work should make a good contribution to the literature. Further more, in my opinion, the paper will attract a wide readership. The MS looks good. Therefore, I have no serious substantive comments to the MS.

Please read the MS  once more and correct any minor shortcomings, e.g. punctuation, etc.

Author Response

Reply to Reviewer 3

Comments and Suggestions for Authors

The article submitted for review meets the standards of a review article. It raises an interesting issue of stress and beneficial rhizobacteria.  Authors  focuses on the current trends of beneficial rhizobacteria toward sustainable enhancement of soil and plant health,

Moreover, all  it is supported very reliably by a large amount of literature.

All tables are presented in a clear way and also figures.

In my opinion this work should make a good contribution to the literature. Further more, in my opinion, the paper will attract a wide readership. The MS looks good. Therefore, I have no serious substantive comments to the MS.

Please read the MS  once more and correct any minor shortcomings, e.g. punctuation, etc.

Reply: Thank you so much for your valuable comments. We have revised the whole manuscript according to your suggestions. Please have a look to our revised manuscript.